# Herd Immunity to SARS-CoV-2 Among the Armenian Population in the Second Half of 2022

**DOI:** 10.3390/epidemiologia6030029

**Published:** 2025-06-20

**Authors:** Anna Yuryevna Popova, Vyacheslav Sergeevich Smirnov, Svetlana Alexandrovna Egorova, Gayane Gurgenovna Melik-Andreasyan, Stepan Armenovich Atoyan, Angelika Marsovna Milichkina, Irina Viktorovna Drozd, Gennady Hovsepovich Palozyan, Valery Andreevich Ivanov, Edward Smith Ramsay, Oyuna Bayarovna Zhimbayeva, Ara Shaenovich Keshishyan, Olga Alexandrovna Petrova, Alexandra Valerievna Gubanova, Alexandra Petrovna Razumovskaya, Anaida Vasilevna Tsakanyan, Armine Varshamovna Margaryan, Tatevik Surenovna Khachatryan, Areg Artemovich Totolian

**Affiliations:** 1Federal Service for the Oversight of Consumer Protection and Welfare (Rospotrebnadzor), Moscow 127994, Russia; depart@gsen.ru; 2Saint Petersburg Pasteur Institute, St. Petersburg 197101, Russia; egorova72@mail.ru (S.A.E.); amilichkina@mail.ru (A.M.M.); ckdl@pasteurorg.ru (I.V.D.); korsaring@yandex.ru (V.A.I.); damaoyuna@rambler.ru (O.B.Z.); belka-mbf1988@mail.ru (O.A.P.); gubanovasasha2012@gmail.com (A.V.G.); sandra020295@gmail.com (A.P.R.); totolian@spbraaci.ru (A.A.T.); 3National Center for Disease Control and Prevention, Yerevan 0025, Armenia; gayamelikandreasyan@gmail.com (G.G.M.-A.); stepan.atoyan@ncdc.am (S.A.A.); ncdc.palozyan@gmail.com (G.H.P.); keshar@mail.ru (A.S.K.); tsakanyananaida@gmail.com (A.V.T.); tatkhach1955@gmail.com (T.S.K.)

**Keywords:** COVID-19, evolution, Armenia, volunteers, herd immunity, seroprevalence, nucleocapsid, receptor-binding domain, vaccination

## Abstract

**Aim.** This study aimed to assess the SARS-CoV-2 herd immunity in the Republic of Armenia (RA) by late 2022. **Materials and Methods.** A randomized study was conducted from 28 November to 2 December (2022) by the Saint Petersburg Pasteur Institute (Russia) in collaboration with the National Center for Disease Control and Prevention (Armenia). This study was approved by the ethics committees at both organizations. A volunteer cohort (N = 2974) was formed and grouped by participant age, region, or activity. Antibodies (Abs) to viral nucleocapsid antigen (Nc) and receptor-binding domain (RBD) in plasma were determined by ELISA. The statistical significance of differences was calculated using a *p* < 0.05 threshold, unless noted. **Results.** At the end of 2022, estimated SARS-CoV-2 seroprevalence (Nc and/or RBD Abs) among the Armenian population was 99% (95%CI: 98.5–99.3). It was evenly distributed throughout the cohort without any significant differences by age, region, or activity. Volunteers with low (32–124 BAU/mL) or medium (125–332 BAU/mL) anti-Nc Ab levels prevailed: 32.4% (95%CI: 30.7–34.1) and 25.5% (95% CI: 24.0–27.1), respectively. Regarding anti-RBD Abs, maximum levels (>450 BAU/mL) were detected in 40% of children. The share of individuals with high anti-RBD Abs levels increased with age, reaching 65% among those aged 70^+^ years. The important contribution to the formation of herd immunity to coronavirus infection was made by vaccination in the preceding period (1 April 2021 to 1 May 2022). The contribution from individuals with post-vaccination immunity was estimated to be above 80%. Hybrid immunity, formed after vaccination of those who had earlier experienced COVID-19, was characterized by greater effectiveness than post-vaccination immunity alone. **Conclusions.** Within the context of mass prophylactic vaccination, effective herd immunity to SARS-CoV-2 was formed, which helped to stop epidemic spread in the Republic.

## 1. Introduction

The 21st century’s first SARS-CoV-2 pandemic quickly became a global challenge. As of 21 January 2023, more than 743 million infections have been registered worldwide, including more than 6.7 million COVID-19 fatalities [1]. Pandemic damage to the economy has been estimated to be at least 3.3% of global GDP [2]. The Republic of Armenia (RA), as a nation, was not significantly involved in the overall pandemic. There were 445,690 confirmed cases in the entire RA, representing about 14% of the Armenian population or 0.06% of those infected globally. There were 8,713 COVID-19 fatalities in the country. Accordingly, the mortality rate was 1.96%, which is 1.98-fold higher than the global average [3]. The SARS-CoV-2 pandemic in the RA started on 3 March 2020, when the first case of COVID-19 was registered. Subsequently, the process proceeded in six waves (Figure 1).

During the 1st peak (weeks 27 to 32, 2020), the Alpha viral variant (B.1.1.7) dominated. During the 2nd peak (weeks 42 to 52, 2020), the Delta variant dominated, and its circulation continued until July 2021. The Omicron variant circulated from the end of 2021 until January 2022 [4]. Subsequently, a continuous change in genetic variants was noted: Kraken (XBB.1.5), Acrux (XBB.2.3), Arcturus (XBB 1.9.1), Hiperion (FL.1.5.1), Fornax (EG.5.1), Eris (BA.2.86.1), and Pirola (JN.1). Despite a wide range of successive SARS-CoV-2 genetic variants, starting from the 40th week (2022), incidence decreased to about 2200 cases per week. By the 46th week of 2023, some stabilization of the number of infections at a level of 2100 cases per week was noted. However, already from the 11th week of 2024, a slight increase to about 5000 cases per week was observed (Figure 1) [1].

Starting from 1 April 2021, a vaccination campaign against SARS-CoV-2 was launched in the RA. In the initial period, the Gam-COVID-Vac vector vaccine was mainly used. Further, the range of vaccines expanded to include AZD122ChAdXo1, BNT162b2, mRNA1273, and Sinopharm BBIP [5]. It is interesting to note that after reaching 30% vaccination coverage (weeks 5–6 of 2022), a decrease in incidence was observed (Figure 1). This turned into a steady downward trend, which was replaced, in essence, by seasonal dynamics from the beginning of 2023. By this time, vaccination coverage reached almost 35%. The overview of local pandemic dynamics we present is completely consistent with previously described epidemiological mechanisms [6]. Among them, upon reaching the threshold level of herd immunity, an epidemic spontaneously dies out or, as in this case, the infection acquires a seasonal course. Hence, it can be reasonably considered that herd immunity is the central factor in protecting a social higher biological species from microbial onslaught.

Herd immunity is a multicomponent complex defense system that includes many cellular and humoral factors. Cellular factors are decisively responsible for the recognition and regulation of functions, and humoral factors are largely responsible for the executive reactions of the immune defense [7,8,9]. An essential part of the latter group are antibodies that have the ability to bind and, in one way or another, inactivate the foreign pathogen [8,9]. The purpose of this study was to assess the humoral herd immunity to SARS-CoV-2 in the Armenian population as of the end of 2022.

## 2. Materials and Methods

This study is part of a multi-stage implementation of scientific cooperation between the countries of Eastern Europe, Transcaucasia, and Central Asia to study herd immunity to SARS-CoV-2 infection under Russian Government Order (dated 18 June 2021, No. 1658-p); Rospotrebnadzor Order (dated 9 September 2021, No. 512); and Armenian Minister of Health Order (dated 31 January 2022, No. 380-A). The organizers and implementers of this study were the Saint Petersburg Pasteur Institute (Rospotrebnadzor of the Russian Federation) and the National Center for Disease Control and Prevention (Ministry of Health of the Republic of Armenia).

### 2.1. Study Design and Organization

A randomized cohort study of herd immunity to SARS-CoV-2 infection was conducted according to a program developed by the Federal Service for the Oversight of Consumer Protection and Welfare (Russian Federation) with the participation of the Saint Petersburg Pasteur Institute, taking into account WHO recommendations.

The organization and conduct of studies of the formed cohort of volunteers were similar to those described previously [10]. Each volunteer, or their legal representative (in the case of children), was familiarized with the goals and conditions of the upcoming study and signed a statement of informed consent. This study was organized in accordance with the provisions of the Declaration of Helsinki. It was approved by the Ethics Committee, National Center for Infectious Diseases (No. 01 dated 17 February 2022), and the Ethics Committee, Saint Petersburg Pasteur Institute (protocol No. 64, dated 26 May 2020).

### 2.2. General Characteristics of the Volunteer Cohort

This study was conducted in Yerevan and all Armenian regions. Based on initial survey results, a cohort of 2974 volunteers was formed, from whom blood samples were taken between 28 November and 2 December 2022. The distribution of volunteers within most adult age brackets turned out to be relatively homogeneous, both in absolute numbers and in proportional representation (Table 1). The ‘1–17 years old’ and ‘18–29 years old’ groups were an exception, and their representation was significantly lower than that of older groups.

Study participants represented 0.1% (95%CI: 0.10–0.11) of the Armenian population (Table 2). As expected, the largest number of volunteers was recruited in the capital, Yerevan (35% of the cohort). However, given the actual population of the city (1,098,900 as of December 2022), the actual representation of volunteers there was 0.1%, which matches other regions. The share of volunteers (percentage of the overall cohort) from other regions varied from 11.1% (Gegharkunik Province) to 2.2% (Vayots Dzor Province).

As in other population-based studies, women participated more actively than men [11]. The participants were 557 male (18.7%; 95%CI: 17.4–20.2) and 2,317 female (81.3%; 95%CI: 79.8–82.6.0), respectively. Among the volunteers, 983 people (33.1%; 95% CI: 31.4–34.8) had a COVID-19 history with confirmation of diagnosis by a medical institution. The cohort included representatives of twelve activity groups, including unemployed, pensioner, preschool and school-age children, and students (Table 3).

The most well-represented group was healthcare workers, whose share in the surveyed cohort was 47.9% (95%CI: 46.1–49.7). The smallest shares (<1.5%) were noted in the categories ‘business’, ‘science and arts’, ‘students’, and ‘production and transport’. In some cases, due to the small number of volunteers in the categories, they were combined: preschoolers with schoolchildren; military personnel with civil servants; office workers with IT specialists, scientists with artists, and industrial workers with transportation workers (Table 3).

### 2.3. Organization of Laboratory Research

The presence and levels of IgG antibodies to two SARS-CoV-2 antigens were assessed in volunteer plasma: Nc and the RBD region of the S protein. A detailed description of the organization of laboratory analysis is given in our previous publication [11]. To obtain comparable results, it was fundamentally important to strictly adhere to the previously described research methodology.

### 2.4. Statistical Analysis

Statistical analysis was performed using Microsoft Excel 2010. Mean values and confidence intervals of the proportions were calculated using the method of A. Wald and J. Wolfowitz [12] with correction according to A. Agresti and B. A. Coull [13]. Pairwise rank correlation according to Spearman was performed using the Statistica 13 package [14]. The statistical significance of differences in proportions was calculated by z-test using a special calculator [14]. For other statistical calculations, Statistica 13 was used. The significance of differences, unless otherwise stated, was assessed with a probability of *p* ≤ 0.05. Graphic illustrations were made in Microsoft Excel 2010.

## 3. Results

### 3.1. SARS-CoV-2 Seroprevalence by Age

By the beginning of this study, the SARS-CoV-2 pandemic had passed the 34-month mark. During this time, 445,690 COVID-19 cases were confirmed in the Republic, and almost 35% of the population was vaccinated [5]. It is worth noting that the total number of asymptomatic and/or undiagnosed cases of COVID-19 in the RA is unknown. Under these conditions, a high level of herd immunity among volunteers would be expected. The results obtained fully confirm the validity of this assumption (Figure 2).

The data shows that the absolute majority of volunteers had Abs to both main antigens. The total share of seropositive individuals (Nc and/or RBD) in the cohort as a whole was 99% (95%CI: 98.5–99.3). The cohort was dominated by individuals seropositive for both Abs (Nc^+^RBD^+^), reaching 88.7% (95%CI: 87.5–89.1). The share of individuals who were only positive for anti-RBD Abs (Nc^−^RBD^+^) was an order of magnitude lower at 8.9% (95%CI: 7.9–10.0) on average. Regarding seronegative (Nc^−^RBD^−^) and those only seropositive for Nc (Nc^+^RBD^−^), their shares did not exceed the standard error calculated for the double-positive (Nc^+^RBD^+^) group. They were 1.3% (95%CI: 0.9–1.8) and 1.0% (95%CI: 0.7–1.5), respectively.

### 3.2. Seroprevalence by Region and Activity

The Republic of Armenia is the smallest state in the Caucasus region (29,743 km^2^). Considering the high level of herd immunity among volunteers in all age groups, and the small geographic area involved, it would be reasonable to assume that region of residence within the RA would not have a significant impact on seroprevalence. Analysis confirmed this assumption. High seroprevalence levels had formed throughout Armenian regions, including about 90% prevalence of double-seropositive status (Nc^+^RBD^+^). This status ranged from 81.2% (95%CI: 69.5–89.9) in the Vayots Dzor region to 92.8% (95%CI: 89.4–95.3) in Gegharkunik.

Occupation is likely a significant factor determining SARS-CoV-2 infection risk and associated seroprevalence. In particular, it is believed that individuals who have frequent professional contacts with the population are most vulnerable to SARS-CoV-2 infection [15,16]. As such, assessing seroprevalence levels among individuals of different professions and activities seemed relevant. No significant differences from the average cohort value were found in any professional group.

As for the structure of immunity, it closely matched the general trends described in the age group analysis. There was a predominance of double-positive (Nc^+^RBD^+^) volunteers in all groups, ranging from 82.4% (95%CI: 69.1–91.6) among business workers to 93.8% (95%CI: 88.2–97.3) among education personnel. There was a smaller, insignificant contribution to seroprevalence from mono-positive RBD status (Nc^−^RBD^+^). The two remaining Nc-related categories (Nc^+^RBD^−^, Nc^−^RBD^−^) were minimal. Thus, SARS-CoV-2 seroprevalence levels among Armenian volunteers were close to maximum values. Herd immunity was not dependent on age, region of residence, or activity.

### 3.3. Quantitative Ab Levels (Nc, RBD) by Age

Based on the results of the qualitative analysis, namely the distribution of seropositivity to the two main viral antigens, it seemed advisable to assess the quantitative structure of Ab distribution by age group. This study showed a predominance of individuals with low (32–124 BAU/mL) or medium (125–332 BAU/mL) Ab levels in all age groups (Figure 3).

When characterizing the observed distribution, it is worth paying attention to the absence of significant differences in anti-Nc Ab levels between groups. Regression analysis indicated an insignificant downward trend in the older age groups, which was most likely random. This is reflected in the low determination coefficient and the absence of any correlation dependence in the distribution. This likely indicates a uniform distribution of anti-Nc Abs in the population in the range 32–124 BAU/mL.

The fewest individuals were in the maximum Ab category (>666 BAU/mL). Within it, a tendency towards a higher share of seropositive individuals with maximum anti-Nc level was noted with age. In two older groups (50–59, 70^+^), the shares of such individuals were significantly higher than in the ‘30–39- years old’ group (*ρ =* 0.78; *p* = 0.05). In contrast, the share of seronegative individuals showed a downward trend from the ‘1–17 years old’ group to the 70^+^ group, but the trend was insignificant (*ρ =* −0.57; *p* > 0.1).

Of some interest may be the distribution of volunteers with plasma anti-Nc Abs within the range 17–31 BAU/mL. In this case, a significant downward trend was noted (*ρ =* −0.89; *p* = 0.01). Considering that a similar trend was noted in the group with anti-Nc titers within 32–124 BAU/mL, it can be concluded that volunteers with very low (17–31 BAU/mL) or low (32–124 BAU/mL) level are most widely represented in the age group 1–17 years. These shares gradually decrease with advancing age until the group ‘70^+^ years’.

The second marker analyzed was antibodies against RBD region of the S protein. Such Abs play an important protective role in the pathogenesis of infection [17]. Assessment of the quantitative distribution of anti-RBD Abs in volunteers of different ages coincided with the previous conclusion about achievement of a high level of herd immunity (Figure 4).

In all the age groups, volunteers with the maximum anti-RBD IgG level (>450 BAU/mL) predominated, and their share increased with age. The trend line is described by a linear regression with an upward trend across all age groups starting from ‘1–17 years’ until ‘70^+^’ years, with a slope (k) equal to 4. The correlation coefficient (*ρ =* 0.94) indicates a strong relationship between age and the share of carriers with the maximum anti-RBD Ab level, and the correlation coefficient in this group was significant (*p* = 0.001). Interestingly, in groups with medium (220.1–450.0 BAU/mL) or low (22.6–220.0 BAU/mL) Ab levels, the trend is reversed, and their slopes also decrease to −1.66 and −1.8, respectively. The *ρ* values become negative, yet the significance of the relationships remains high (0.01 and 0.001, respectively). The share of individuals with very low anti-RBD Ab levels remained virtually at a minimum and did not depend on age.

In summarizing the data, we can make a general conclusion that the quantitative distribution of SARS-CoV-2 Abs in the COVID-19 post-pandemic period reflects some specificity. This is manifested as a high share of individuals with low or moderate anti-Nc Ab levels in the context of a high share of volunteers with the maximum level of anti-RBD Abs.

### 3.4. Volunteer Vaccination

Vaccination played an important role in stopping the epidemic process of SARS-CoV-2 infection in the RA. By the late January–early February timeframe (2022), in other words before the study period, total vaccination coverage of the local population with COVID-19 vaccines exceeded 60% according to official data. Comparison of infection and vaccination dynamics showed a significant decrease in infection rate from the moment the aforementioned threshold was reached. Thus, starting from week 4, 2022, a stable inverse correlation was revealed between the increase in vaccination coverage and the rate of decline in the number of infections (Figure 1). The Spearman correlation coefficient (*ρ*) for the time period from week 4 to week 44 (2022) was −0.44 (*p* < 0.001).

Targeted and consistent COVID-19 vaccination was implemented. A sizeable portion of the population also experienced infection, either in a manifest or asymptomatic form. Together, these naturally led to the cessation of the pandemic process through achievement of almost ideal herd immunity. This was true in almost all Armenian age groups (Figure 5), and most groups featured maximal anti-RBD Ab levels (Figure 4).

A wide range of vaccines was used in the RA in 2021. We have described the logistics of their usage in detail in a previous report [18]. By the beginning of 2022, this campaign had already been completed in most of the Republic, which was reflected as a natural decrease in the rate of vaccination (Figure 5).

According to online data [5], the maximum number of people who fully completed SARS-CoV-2 vaccination occurred in the autumn of 2021. Afterwards, immunization activity consistently decreased, and it almost completely ended in April–May 2022, which is when COVID-19 incidence reached a sporadic level. At the end of 2022 during the study period, vaccination coverage in the surveyed volunteer cohort exceeded 82.9% (95%CI: 81.5–84.2) in all age groups, with the exception of children (15.5%; 95%CI: 11.6–20.4) (Figure 6).

To achieve greater representativeness, all vaccines approved in the Republic were grouped into three main categories based on production platform. The ‘vector’ group combined GamCOVIDVac (Russia) and AZD1222ChAdO1 (UK). The ‘mRNA’ group combined the BNT 162b2Vaccine (USA) and the mRNA1273 COVID-19 Vaccine (USA). The “whole virion” group combined CoviVac (Russia), CoronoVac (China), and Sinopharm BIBP (China) (Figure 7).

Whole-virion (43.5%; 95%CI: 41.6–45.4) and vector (42.3%; 95%CI: 40.4–44.2) vaccines were most often used for immunization (Figure 7). This trend was noted in all age groups. The share of mRNA vaccines was 14.2% (95%CI: 13.3–15.6), and vaccines from this platform were more often used in children and young adults (Figure 8).

This study compared the efficacy of hybrid and post-vaccination immunity in Armenian volunteers. For this purpose, COVID-19 incidence (laboratory-confirmed) was compared in two groups of vaccinated volunteers. Among those with only post-vaccination immunity (could not recall symptomatic illness during anamnesis), 17.0% (95%CI: 14.7–19.6) fell ill after vaccination, regardless of the vaccine type used. Among individuals with hybrid immunity (vaccinated after manifest COVID-19), 1.5% (95%CI: 0.6–3.5) fell ill again after vaccination. For comparison, among unvaccinated volunteers, 28.0% (95%CI: 24.4–32.0) fell ill with manifest, laboratory-confirmed coronavirus infection (Table 4). Thus, among individuals with hybrid immunity, COVID-19 incidence was 11-fold lower than among those with post-vaccination immunity and almost 19-fold lower than among those without immunity.

In addition, quantitative indicators of immunity (anti-RBD IgG levels) were assessed in the two groups immunized with vaccine types (vector, mRNA) that initiate production of Abs to the RBD region of the S protein SARS-CoV-2. History of infection was assessed by the presence or absence of anti-Nc Abs insofar as the aforementioned vaccine types do not elicit those specific antibodies. A group of 27 volunteers had post-vaccination immunity alone, as inferred from other factors: no recall of symptomatic illness as well as an absence of anti-Nc Abs. A group of 291 individuals had Abs to both RBD and Nc, indicating a hybrid immune response formed as a result of infection and vaccination (Figure 9).

When comparing volunteers with hybrid immunity with those with post-vaccination immunity, the share of those with low anti-RBD level was lower (3.1%; 95%CI: 1.6–5.8 and 22.2%; 95%CI: 10.6–40.8, respectively). In the same comparison (hybrid, post-vaccination), the share of individuals with high anti-RBD level (>450 BAU/mL) was higher (68.7%; 95%CI: 63.2–73.8 and 44.4%; 95%CI: 27.6–62.7, respectively).

Thus, hybrid immunity (formed after vaccination in volunteers who had previously experienced COVID-19) was characterized by greater efficacy in protecting against symptomatic infection. It featured higher anti-RBD Ab levels compared to ‘pure’ post-vaccination immunity.

## 4. Discussion

Having started on 31 December 2019, the SARS-CoV-2 pandemic evolved over the next 48 months from a global pandemic affecting almost all countries to a seasonal illness with a relatively favorable course. Worldwide, cases exceeded 700 million, including almost 7 million COVID-19 fatalities. Additionally, in tens of millions of cases, illness transformed into post-COVID syndrome, also termed ‘long COVID’ [19,20]. In Armenia, the SARS-CoV-2 pandemic involved more than 450,000 infections, and approximately 8700 individuals died with a diagnosis of COVID-19 [1].

The most important event in the history of the pandemic was the development of a family of anti-COVID vaccines [21]. Widespread and, most importantly, large-scale usage of the developed vaccines was an essential component of a set of anti-epidemic measures. The vaccination process in the RA started on the 14th week of 2021. By January 2022, immunized individuals exceeded 25%. As a result of vaccination and infection, the total herd immunity formed by the specified time at the 99% level (95% CI: 98.5–99.3). Within this number, the largest share of volunteers (88.7%; 95%CI: 87.5–89.1) had both Abs simultaneously (Nc^+^RBD^+^). The share of RBD^+^ mono-positive volunteers was approximately an order of magnitude smaller on average (8.9%; 95%CI: 7.9–10.0), without any significant differences between age groups. Finally, the shares of Nc^+^ mono-positive and completely seronegative (Nc^−^RBD^−^) volunteers varied within statistical error. Age, region, and activity-based seroprevalence distributions were fairly uniform without any significant differences.

Comparison of the quantitative and qualitative distribution of specific plasma Abs allowed us to further characterize the structure of herd immunity. The share of volunteers with low or medium anti-Nc Ab levels (32–124 BAU/mL, 125–332 BAU/mL) reached 32.4% (95%CI: 30.7–34.1) and 25.5% (95%CI: 24.0–27.1), which corresponded with the majority of Nc^+^ individuals, respectively. The share of those with the maximum level of RBD Abs (>450 BAU/mL) rose across age groups from 41.4% (95%CI: 35.2–46.6) among ’children aged 1–17 years’ to 64.8% (95%CI: 59.9–69.5) in the oldest interval. The trend for increasing RBD Ab levels was linear (slope = 4.4).

An important component of resistance to COVID-19 is post-vaccination immunity. The bulk of vaccination in the RA was carried out in the period from April 2021 to May 2022. During this period, a wide range of specific vaccines was used. Vector (Gam-COVID-Vac, AZD122ChAdOxq) and whole-virion (CoviVac, CoronaVac, Sinopharm BBIP) vaccines were used with approximately equal frequency. The share of mRNA vaccines did not exceed 15%. By the end of 2022, vaccination coverage (≥1 dose) in the surveyed cohort exceeded 80% in all age groups except children. The rate of vaccination has decreased to a minimal level. As a result of vaccination and infection, seropositive status (excluding Nc^+^RBD^−^ ‘mono-positive’) reached 97.6% (95%CI: 97.0–98.1).

Hybrid immunity formed after vaccination in volunteers who had previously experienced COVID-19 was characterized by high efficacy in protecting against clinically apparent infection, and it featured high levels of produced antibodies. Herd immunity to SARS-CoV-2 consists of two components. The first, natural immunity, is formed as a result of infection of a susceptible organism with a pathogenic agent causing manifest or asymptomatic infection. The second, artificial (post-vaccination) immunity, is an immune response to the use of specific vaccines. It is believed that primary immunity caused by the original SARS-CoV-2 strain is formed by Nc-specific IgG in 75% of those who have recovered, while vaccination is accompanied by the production of mainly anti-RBD Abs [22].

During the first 4–6 months after infection, a decrease in anti-Nc Ab levels is observed, followed by stabilization over the next 12 months in approximately 80% of those who have recovered [23]. Such reduced Ab levels may be present when an individual has contact with a new SARS-CoV-2 genetic variant, potentially leading to a breakthrough infection [24]. COVID-19 relapses might not be numerous, but their significance should not be underestimated. A rational approach to preventing relapse, from the point of view of classical immunology, could be post-infection vaccination. The emergence of a family of SARS-CoV-2 vaccines designed on different platforms helps implement this idea, and several were used in the examined volunteer cohort (Figure 7).

It is believed that vaccination of those who have recovered from manifest COVID-19 reduces the risk of relapse by about half [25]. It has been shown that vaccination after an infection is accompanied by a significant increase in anti-RBD Ab levels, the decrease in which stabilizes at 6–9 weeks after the 2nd injection [26]. The observed effect was first described by A. Sette and D. Crotty as hybrid immunity [27]. Initially, hybrid immunity was understood as the reactivity formed in COVID-19 convalescents who later received vaccination against SARS-CoV-2 and only in this sequence. However, it soon became clear that the sequence of infection and vaccination does not matter, and hybrid immunity is formed in any case [28].

Interestingly, longer intervals between infection and vaccination provide higher and even cross-protection, and the sequence of events does not affect the production of antibodies [29]. Thus, hybrid immunity appears to be the optimal system for the formation of immune resistance, especially with circulation of such a variable virus as SARS-CoV-2, since it provides an immune response with the greatest breadth and longevity. The results here indicate that in the processes of epidemic viral spread (as COVID-19 or asymptomatic infection) and further active vaccination of the Armenian population, strong herd immunity was formed which helped stop the spread of illness [30].

### Limitations of This Study

The authors would like to note several factors that might affect the sample representativeness or conclusions reached through data analysis. Despite the fact that an information campaign was carried out as widely as possible for the population (state television channels, news sites, Facebook), limited Internet access is a problem for parts of the rural population. Furthermore, residents who are more involved with their health and that of their loved ones (primarily women and healthcare workers) are more likely to take part in studies of this kind. Healthcare workers are well motivated, partially due to frequent contacts with COVID-19 patients. Since this study included an assessment of post-vaccination seropositivity, vaccinated residents were also more interested in participating. These factors may form a bias toward greater representation of women, healthcare workers, and vaccinated residents.

Due to the small number of compared volunteers who had a well-documented history (illness, vaccination), further division of the volunteer comparison groups reduces analytic reliability. The groups are already small. As such, it was not possible to conduct a multivariate analysis, such as with age or vaccine type.

The design of the population study involved screening for antibodies in a large cohort of volunteers, so the ELISA method was used. The authors understand that binding antibody titers may not directly correlate with protective immunity, especially against emerging variants, and indeed this may be viewed as a potential limitation. Unfortunately, the method using the neutralization reaction is not suitable for large-scale studies.

## 5. Conclusions

1. COVID-19 infection, which began in 2020 in the RA, evolved into an acute respiratory, predominantly self-limiting, infection with a seasonal course.

2. By the end of 2022, SARS-CoV-2 herd immunity had reached 99%, and the majority of volunteers (88.7%) had antibodies to both Nc and RBD.

3. The main contribution to limiting epidemic viral incidence was made by hybrid immunity. This was formed due to vaccination of the population, including many who had also recovered from natural infection earlier (2021–2022). The share of individuals who fully completed vaccination exceeded 80% in all cohort age groups, with the exception of children.

4. Hybrid immunity is characterized by high antibody levels and robust protection against symptomatic infection.

## Figures and Tables

**Figure 1 epidemiologia-06-00029-f001:**
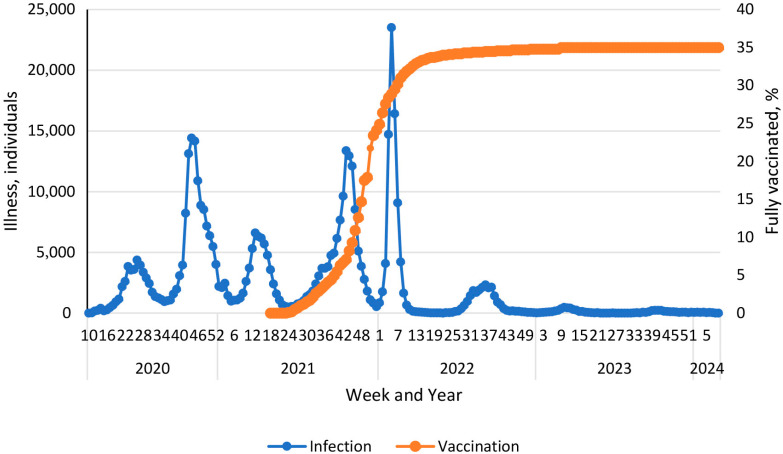
Vaccination and morbidity dynamics during the COVID-19 pandemic in the Armenian population [1].

**Figure 2 epidemiologia-06-00029-f002:**
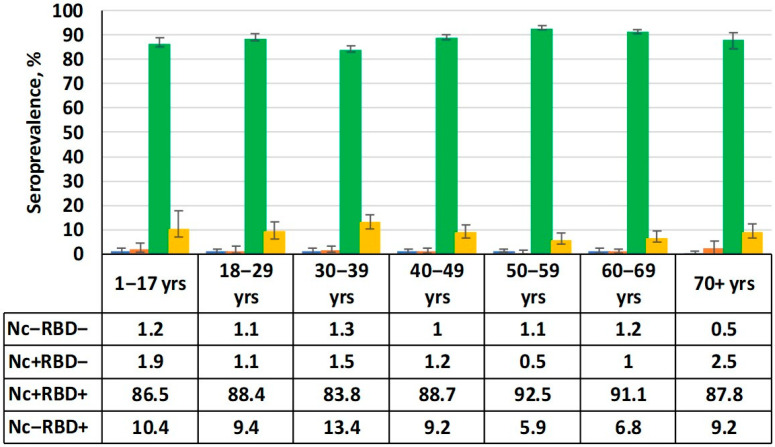
Serological status by age group. Numerical values are given in the table below the figure. Black vertical lines are 95% confidence intervals. Note: Nc^−^RBD^−^ is seronegative; Nc^+^RBD^−^ is seropositive (anti-Nc only); Nc^+^RBD^+^ is double-seropositive (anti-Nc, anti-RBD); Nc^−^RBD^+^ is seropositive (anti-RBD only).

**Figure 3 epidemiologia-06-00029-f003:**
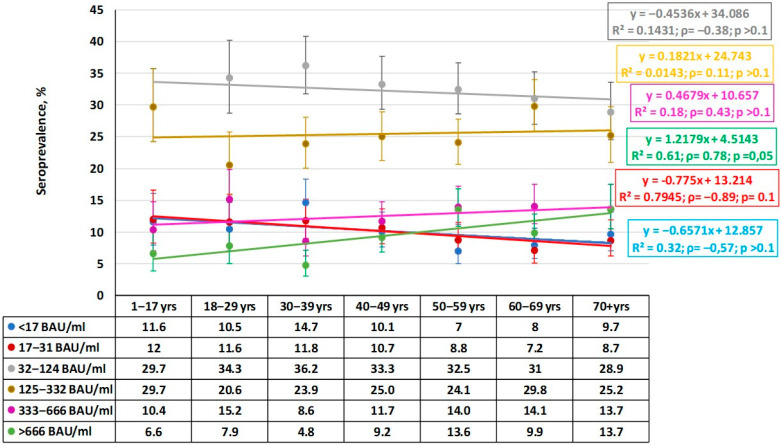
Quantitative distribution of plasma anti-Nc IgG by age group. Legend: Ab ranges are in BAU/mL. Colored lines are trends for each range. Regression equations, determination coefficients, and other values (*ρ*, *p*) are given to the right. Trend lines and statistical information are color matched to the titer range. Vertical black lines are 95% confidence intervals. Numerical values are given in Appendix A.

**Figure 4 epidemiologia-06-00029-f004:**
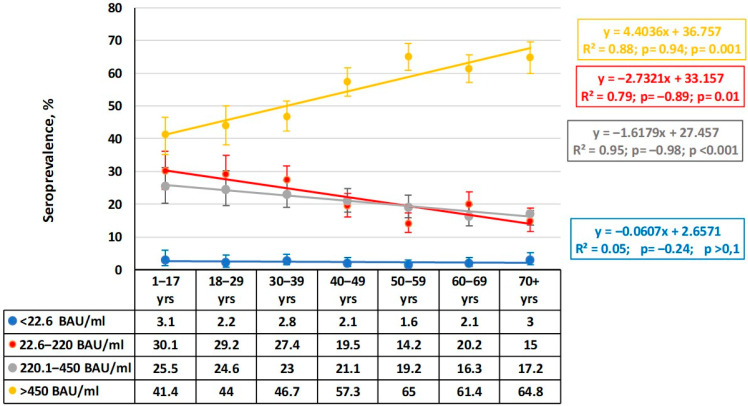
Quantitative distribution of plasma anti-RBD IgG by age group. Legend: Ab ranges are in BAU/mL. Colored lines are trends for each range. Regression equations, determination coefficients, and other values (*ρ*, *p*) are given to the right. Trend lines and statistical information are color matched to the titer range. Vertical black lines are 95% confidence intervals. Numerical values are given in Appendix A.

**Figure 5 epidemiologia-06-00029-f005:**
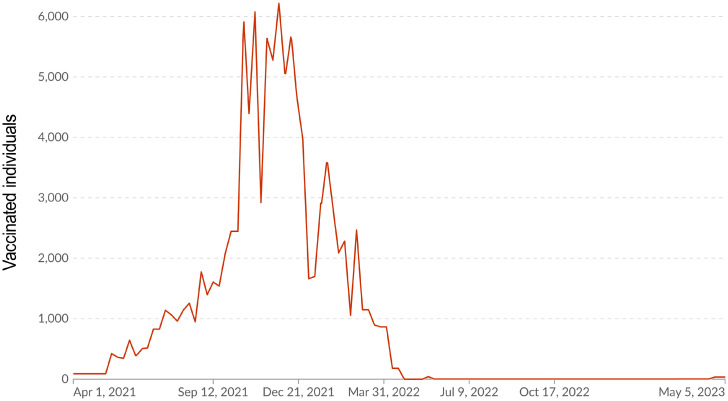
COVID-19 vaccination dynamics. The x-axis shows date of registration. The y-axis shows individuals who completed vaccination. Source: online portal [5].

**Figure 6 epidemiologia-06-00029-f006:**
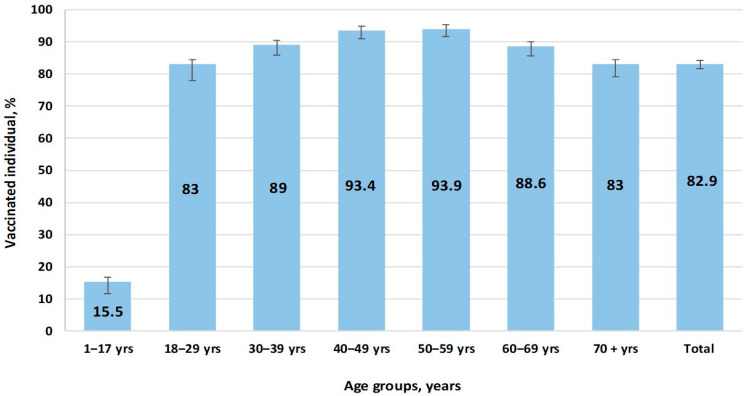
Vaccination coverage by age group.

**Figure 7 epidemiologia-06-00029-f007:**
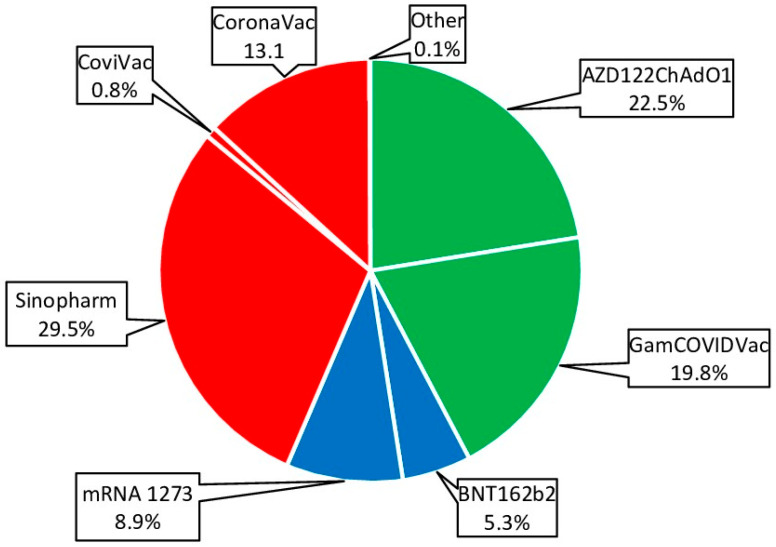
Vaccines used for specific prevention of COVID-19. Percentages are calculated as share of all vaccinations. Legend: whole-virion vaccines are highlighted in red; vector vaccines in green; mRNA vaccines in blue.

**Figure 8 epidemiologia-06-00029-f008:**
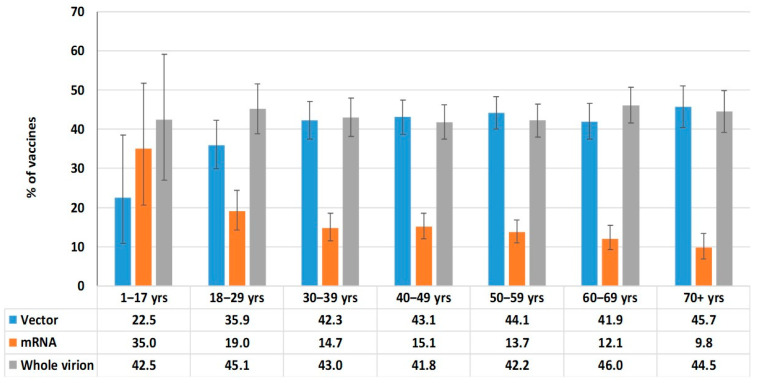
Vaccine categories used by age group.

**Figure 9 epidemiologia-06-00029-f009:**
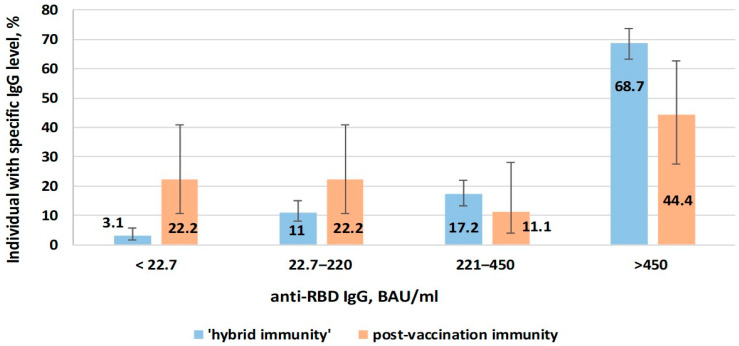
Anti-RBD IgG levels in volunteers vaccinated with vector or mRNA vaccines depending on type of immunity.

**Table 1 epidemiologia-06-00029-t001:** Age structure of the volunteer cohort.

Age Group, Years		Volunteers		
		n	Share, %	95% CI
**1–17**		258	8.7	7.7–9.7
* subgroup	1–6	21	8.1	5.1–12.2
	7–13	81	31.4	25.8–37.4
	14–17	156	60.5	54.2–66.5
18–29		276	9.3	6.7–10.4
30–39		456	15.3	14.1–16.7
40–49		512	17.2	15.9–18.6
50–59		556	18.7	17.3–20.1
60–69		516	17.4	16.0–18.8
70^+^		400	13.4	12.3–13.7
total:		2974	100	

Note: * subgroups within volunteers aged 1–17 years; 70^+^–70 years and older.

**Table 2 epidemiologia-06-00029-t002:** Distribution of volunteers by region.

Region	n	Share, %	95% CI
Yerevan	1041	34.7	33.0–36.4
Gegharkunik Province	330	11.1	10.0–12.3
Armavir Province	265	8.9	7.9–10.0
Lori Province	259	8.7	7.7–9.8
Kotayk Province	247	8.3	7.3–9.4
Shirak Province	248	8.3	5.0–9.4
Ararat Province	164	5.5	4.7–6.4
Tavush Province	155	5.2	4.5–6.1
Syunik Province	122	4.1	3.4–4.9
Aragatsotn Province	79	2.7	2.1–3.3
Vayots Dzor Province	64	2.1	1.7–2.7
Total:	2974	100	

**Table 3 epidemiologia-06-00029-t003:** Volunteer distribution by activity.

Activity	n	Share, %	95% CI
healthcare	1425	47.9	46.1–49.7
unemployed	357	12.0	10.9–13.2
pensioner	308	10.4	8.9–11.5
other	200	6.7	5.7–7.7
preschool or schoolchild	193	6.5	5.6–7.4
education	129	4.3	3.6–5.1
state military service	92	3.1	2.5–3.8
I.T. and office work	85	2.9	2.3–3.5
business	51	1.7	1.2–2.2
science and the arts	50	1.7	1.2–2.2
student	49	1.6	1.2–2.2
industry and transportation	35	1.2	0.8–1.6
total:	2974	100	

**Table 4 epidemiologia-06-00029-t004:** SARS-CoV-2 infections following immunization by vaccine type. The data reflect volunteers (n = 1258) who contracted COVID-19 after vaccination.

Vaccines	Laboratory-Confirmed COVID-19 Cases Following Vaccination							
	Volunteers with Post-Vaccination Immunity ^1^				Volunteers with ‘Hybrid Immunity’ ^2^			
	N	n	%	95% C.I.	N	n	%	95% C.I.
1. RBD vaccines	387	69	17.8	14.3–22.0	119	1	0.8	0.1–4.6
1.1. Vector vaccines	331	62	18.7	14.9–23.3	86	1	1.2	0.2–6.3
AZD122ChAdOx1	85	9	10.6	5.7–18.9	31	1	3.2	0.6–16.2
Gam-COVID-Vac	246	53	21.5	16.9–27.1	55	0	0.0	0.0–0.0
1.2. mRNA vaccines	56	7	12.5	6.2–23.6	33	0	0.0	0.0–0.0
mRNA1273	52	6	11.5	5.4–23.0	28	0	0.0	0.0–0.0
BNT162b2	4	1	25.0	4.6–69.9	5	0	0.0	0.0–0.0
2. Whole-virion vaccines	541	89	16.5	13.6–19.8	211	4	1.9	0.7–4.8
CoronaVac	179	43	24.0	18.4–30.8	55	0	0.0	0.0–0.0
Sinopharm BBIP	354	43	12.1	9.1–16.0	153	4	2.6	1.0–6.5
CoviVac	8	3	37.5	13.7–69.4	3	0	0.0	0.0–0.0
Total:	928	158	17.0	14.7–19.6	330	5	1.5	0.6–3.5

^1^ The volunteer denied a history of manifest COVID-19 before vaccination. ^2^ The volunteer experienced manifest, laboratory-confirmed COVID-19 before vaccination. N is the number of vaccinated individuals, and n is the number who fell ill after vaccination.

## Data Availability

The data are contained within the article.

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
