# Peer review of "Herd Immunity to SARS-CoV-2 Among the Armenian Population in the Second Half of 2022"

_epidemiologia, 2025, doi:10.3390/epidemiologia6030029_

Round 1
Reviewer 1 Report
Comments and Suggestions for Authors
The paper is interesting, but a little long. About innate immunity I suggest this paper:
Palmieri B, Manenti A, Galotti F, Vadalà M. Innate Immunity Stimulation during the COVID-19 Pandemic: Challenge by Parvulan. J Immunol Res. 2022 Apr 29;2022:4593598. doi: 10.1155/2022/4593598. PMID: 35528612; PMCID: PMC9076318.
Author Response
All responses to Reviewer are in the attached file

Reviewer 2 Report
Comments and Suggestions for Authors
This study aims to assess herd immunity against SARS-CoV-2 in the Republic of Armenia.
There are two points that the reviewer does not understand:
There are many papers stating that the incidence of C-19 differs between men and women (gender difference). However, in this paper, men and women were studied together. I would like the validity of this to be demonstrated.
In Western countries, mRNA vaccines are overwhelmingly administered, and this study describes an unfamiliar vaccine. In other words, the effectiveness of a vaccine that is unknown to Western researchers is unclear. I would like this point to be clarified as well.
I think that the rest of the research has been very well done.
Author Response
All responses are in the attached file

Reviewer 3 Report
Comments and Suggestions for Authors
The manuscript focuses on collective immunity in the Republic of Armenia, specifically in the second half of 2022. The paper is clearly structured and contains important information on the spread of antibodies on a cohort of volunteers after the implementation of the collective orders to study collective immunity to countries of Eastern Europe, Transcaucasia and Central Asia. The study involves a randomized cohort of 2,974 volunteers, evaluating anti-nucleocapsid (Nc) and anti-receptor-binding domain (RBD) antibodies via ELISA. The authors report a high seroprevalence (99%) and conclude that hybrid immunity (post-infection plus vaccination) plays a major role in epidemic control. I have the following comments:
- The manuscript mentions low reinfection rates but lacks a detailed breakdown of temporal dynamics.
- While the study reports hybrid immunity superiority, multivariate analysis adjusting for confounders, such as age or vaccine type, would enhance the study. If this is not possible, please add it as a potential limitation.
- While anti-Nc and anti-RBD IgG levels are measured, no neutralization assays were performed. Binding antibody titers may not directly correlate with protective immunity, especially against emerging variants and may be viewed as a potential limitation.
- Along with the mentioned sex, job and vaccination status, there may still be some regional and SARS-CoV-2 variant differences, which may be considered potential limitations. Also, by its nature this study is cross sectional, which has its own, inherent limitations.
- I think the limitation section itself should be at the end of the discussion chapter, rather than after conclusions.
- Minor English improvements: use either "SARS-CoV-2" or "coronavirus" consistently; missing articles here and there
Author Response
All responses to the Reviewer are in the attached file
